# The role of maternal health care services as predictors of time to modern contraceptive use after childbirth in Northwest Ethiopia: Application of the shared frailty survival analysis

**Amanu Aragaw Emiru** [1]*, **Getu Degu Alene**[2], **Gurmesa Tura Debelew**[3]

**1** Department of Reproductive Health and Population Studies, College of Medicine and Health Sciences, Bahir Dar University, Bahir Dar, Ethiopia, **2** Department of Epidemiology and Biostatistics, College of Medicine and Health Sciences, Bahir Dar University, Bahir Dar, Ethiopia, **3** Department of Population and Family Health, Faculty of Public Health, Institute of Health, Jimma University, Jimma, Ethiopia

* amanuaragaw@yahoo.com

**Data Availability Statement:** All relevant data are within the manuscript and its Supporting Information files.

## Abstract

### Introduction

The first year after birth is an ideal time to offer contraception services, as many women have many opportunities to be in contact with the health care system. Nevertheless, a large number of postpartum women in developing countries do not use the service owing to the interplay of factors operating at various stages. Therefore, this study aimed to assess predictors of modern contraceptive use in the extended postpartum period.

### Methods

A community based retrospective cross-sectional study was done among 1281 women who gave birth within 12 months preceding the survey. Kaplan-Meier plots and log rank tests were used to explore the rate of modern contraceptive use. The Weibull regression survival model with multivariate frailty was employed to identify the predictors of time to contraception.

### Results

Of the respondents, 59.1% (95% CI: 56.8%–62.2%) had started using modern contraceptive methods within 12 months after birth. By the second month after birth, only 11.1 percent of the women surveyed started to use a contraceptive method, which increased steadily to 25.9%, 37.7%, and 59.5% at 6, 9, and 12 months, respectively. The most preferred contraceptive method was injectable (71.5%), followed by implants (21.5%). Women's education (aHR = 1.29; 95%CI: 1.02, 1.66), four or more antenatal care (aHR = 1.59; 95% CI: 1.22, 2.06), early initiation of antenatal care (aHR = 2.03; 95% CI: 1.28, 3.21), and early postnatal checkup (aHR = 1.39; 95% CI: 1.12, 1.73) were statistically significant predictors of earlier initiation of modern contraceptive methods.

**Funding:** This manuscript was based on the PhD thesis on the continuum of maternal care, which was conducted with the financial support of Bahir Dar University (URL: https://www.bdu.edu.et). Amanu Aragaw received the award (Research project code No: RCS/252/09). The funder had no role in study design, data collection and analysis, decision to publish, or preparation of the manuscript.

**Competing interests:** The authors have declared that no competing interests exist.

## Conclusions

A substantial proportion of women did not use modern contraceptive methods in the first year after birth. Maternal services were found to be the sole predictors in postpartum contraceptive use. Findings suggest the importance of linking postpartum family planning along the continuum of care. The observed heterogeneity at cluster level also urges the need of disaggregating data for decision-making.

## Introduction

At the turn of the 21$^{st}$ century maternal mortality remains a major public health challenge for many developing countries, and nowhere are global inequalities more starkly clear than in maternal death[1]. The low-income countries accounted for 99% of these deaths of which two-third occurs in Sub-Saharan Africa where Ethiopia lies [1, 2]. At 412 deaths per 100,000 live births[3], Ethiopia's maternal mortality ratio is one of the highest even by the standard of developing countries [4].

However, it is known that many of the maternal deaths can be prevented with appropriate maternal care during pregnancy, delivery and post natal periods [5, 6]. With this in view, women access health services more often during the time of pregnancy, delivery, and the first year after birth than other periods [7].These points of contact, whether part of a routine or emergency care, are opportunities for providers to screen for, counsel, and hence to address the contraceptive needs of postpartum women and couples[8]. Therefore, promoting contraceptive use during each contact in the continuum is considered an important strategy for addressing the widespread unmet needs in family planning [9].

Despite this, the vast majority of postpartum women in developing countries missed the opportunities at each points of contacts [8, 10]. With this, postpartum women are among those with the greatest unmet need for family planning than other periods [6, 10, 11]. In Ethiopia, unmet need for family planning in the postpartum period is also much higher than women outside of the extended postpartum period; while 16.2% women, in general, do have unmet need for family planning[12], the unmet need in the first year after childbirth reaches as high as 44 percent [13].

Women's decision to use modern contraceptives in the first year of postpartum is influenced by a complex array of factors within the community and health system[14, 15]. Postpartum women often do not realize that they are at risk of pregnancy when they are amenorrheic or breastfeeding [15]. Postpartum contraception adoption has also a social connotation; the postpartum period is the time during which a woman needs to adjust herself to new roles to care for her newborn [16]. Furthermore, male dominance and subordination of women, as well as mistaken beliefs, and religious faith of the population are other impediments to the acceptance of contraception in developing countries, including Ethiopia [15].

Beyond the socio-cultural barriers, insufficient contraceptive method mixes, and poor family planning service integration with other health services in many settings also appear to be formidable obstacles remaining to be correct [8, 17].

Several studies have been carried out on postpartum contraceptive use in the study area in particular and Ethiopia at large [18–20]; yet, most of the earlier studies, if any, have ignored the hierarchical facts. It is, however, evidenced that clustering (frailty) has an effect on modeling the predictors of time to contraception[21]. Consequently, modeling time to contraceptive

use ignoring the frailty terms may lead to biased estimates of parameters and their respective standard errors[21, 22].

Furthermore, according to the current Ethiopian Demographic and health survey (EDHS), regional disparities in contraceptive adoption have been reported, with higher rates observed in the Amhara region(where the study area is found) and Addis Ababa(the capital city of the country) than the rest of the regions[3]. Nevertheless, this was a survey report rather than an empirical study, and the contribution of explanatory factors was not examined.

In this regard, addressing these gaps would have significant implications for policymakers, health planners, and clinicians. Therefore, considering the hierarchal nature of our data we attempted to provide an understanding of factors associated with the timing of postpartum contraceptive use using a shared frailty survival model.

## Materials and methods

### Study setting

The study was carried out in West Gojjam zone, which is one of the eleven zones found in the Amhara region of Ethiopia. Administratively, the zone is subdivided into 13 rural districts and 2 town administrations with a projected total population of 2,611,925 (2,194,017 rural and 417,908 urban) people. The number of females in the reproductive age group was 615,892, accounting for 23.58% of the total population[23].

The zone had over 598 health facilities (6 public primary hospitals, 103 health centers, 374 functional health posts, 114 private clinics, and 1 private hospital) at the time of the survey. Family planning, antenatal care, labor and delivery, and postnatal services are provided free of charge in all the public health facilities[23].

### Study design and population

A community-based retrospective cross-sectional study was conducted on reproductive-aged women whose most recent birth was within 12months preceding the survey.

### Sample size and sampling procedure

The required sample size was done through the STAT CALC program of the Epi-Info statistical package V.7.0. This study was part of a large study done on the continuum of maternal health care with multiple objectives. For each objective alternative sample sizes were computed considering both the double and single population formulas; the detail of the sample size calculation and sampling procedure is publicly posted in the research square. Of the alternative sample sizes computed based on different indicators, the largest sample size (1294 women) was obtained when considering the following assumptions; 95% confidence level, 4%margin of error, 16.5% proportion of PNC utilization[24], design effect of 2, and 10% non-response rate. However, during the time of data collection 1337 women who met the inclusion criteria were included in this study to increase the power of the study.

A multistage sampling technique was used to identify the study participants. First, five out of fifteen districts in the Zone (four rural districts and one town administration) were selected using simple random sampling. Second, thirteen kebeles (the smallest administrative units in Ethiopia) were chosen randomly by taking in to account the number of kebeles in each district. Then, a complete list of deliveries that took place within 12 months before the survey was identified from the family folder of health extension workers in the respective kebeles. Finally, 1337 eligible women who met the inclusion criteria were selected.

## Study Variables and measurement

The outcome variable was modern contraceptive use within 12 months following the last birth. A woman who started using modern contraceptives was coded as "1", and otherwise "0".

The explanatory variables included: socioeconomic variables (such as place of residence, maternal/paternal education, household wealth index, and primary Health care (PHC) facilities per 25,000 populations at district level); demographic characteristics (age of women, occupation of women and husbands); and reproductive variables (such as desirability of the pregnancy, family size, birth interval, number and timing of antenatal visits, mode of delivery, and postnatal care.

The wealth index was generated from the household's cumulative living standard based on ownership of specified assets using factor analysis and was later categorized into terciles (poor, middle and rich).

The two quantitative terms, survivor function S (t) and hazard function h (t), are important in any survival analysis[21]. In relation to our study, the survivor function is the probability that a postpartum woman "survives" longer than some specified time "t" without started taking modern contraceptive methods after childbirth. Whereas, the hazard function gives the instantaneous potential per unit time to start using modern contraception after time "t", given that the woman had not started taking any modern contraceptive up to time "t".

## Data collection process

The household data were collected using a pre-tested interviewer-administered questionnaire, developed in the local language, Amharic. Fifteen nurses and five public health officers were deployed as data collectors and supervisors, respectively after receiving two days of intensive training. Data regarding socio-economic, demographic, and reproductive characteristics were collected among women who gave birth (either at home or in a health facility) to a baby within 12 months before the survey. Besides, the number of PHC providing maternity and reproductive health services per total population was assessed at the district level, and the result had been linked to the individual woman in the corresponding household survey.

## Data processing and analysis

The analysis was done using STATA 14.0. Both descriptive statistics and survival analysis techniques were used in analyzing the data. First, an assessment of the time-to-modern contraceptive use after birth was done using life tables based on the Kaplan-Meier (K-M) estimate. Second, the Log-rank Chi-square test was used to examine the differences in the survival curves for different categories of each study variable. Third, the multivariate (or shared frailty) survival analysis was done by assuming different parametric distributions for the baseline hazard function and using gamma for frailty distributions.

The shared frailty approach is a conditional independence model for time to event data, where the frailty term (the random effect) is common to all subjects in a cluster [21]. In our study women who were living in the same cluster (kebele) were more likely to have outcomes (post-partum contraceptive use) that are correlated with one another, and, thus independence between event times cannot be assumed. Moreover, it is unlikely to include all the relevant covariates in the model. With that in mind, using the cox proportional hazard model could not account for all the variability in the observed failure times. Therefore, it was reasonable to apply the shared frailty survival model, that accounts for heterogeneity caused by unmeasured covariates, as an alternative to the standard cox survival model [21, 22]. The conditional hazard

function for the Weibull shared frailty survival model used in this study is defined as:

$$h_{ij}(t/z_i) = z_i \rho t^{\rho-1} \exp(\beta' X_{ij}), i = 1, \ldots, n; j = 1, \ldots, k_i$$

Where $i$ indicates the $i^{th}$ cluster (kebele), $j$ indicates the $j^{th}$ individual in the $i^{th}$ cluster, $pt^{p-1}$ is the baseline hazard, $X_{ij}$ is the vector of covariates for subject $j$ in cluster $i$, $\beta'$ is a vector of regression coefficients, and $Z_i$ is the frailty term. In this study the frailty $Z_i$ was supposed to follow a gamma distribution $g(z; \theta, \theta)$, which is the most common and widely used distribution for determining the frailty effect[22].

The Akaike Information Criterion (AIC) was used to select the appropriate model, whilst the Cox-Snell residual plot analysis was done to evaluate the overall model fitness. Furthermore, interaction between the independent variables for contraceptive use was tested. Finally, the frailty effects, Kendall's Tau, and hazard ratio at 95% confidence interval were estimated for the selected model.

### Ethical approval

This study was approved by the Research Ethical review committee of the College of Medicine and Health Sciences, Bahir Dar University (reference number: 087/18-04). Letters of permission were secured from the Amhara Regional State Health Bureau and respective health offices. Also, oral informed consent was received from each study participant. The data obtained from each study participants was kept confidential throughout the process of study, and the name of the participants was replaced by code.

## Results

### Background characteristics

A total of 1281 reproductive- age women participated in this study giving a response rate of 95.8%. More than half 674(52.6%) of the women were in the age group of 25–34 years with a mean (± SD) age of 30.3(±6.0) years. Little more than three-quarters, 978(76.3%), of the sampled women were rural residents, and 862(67.3%) of them belonged to the lower two wealth quintiles.

Concerning the reproductive characteristics, 511(39.9%) of them had at least 4 ANC visits, 194(15.1%) had their first ANC visit within the first four months after conception. Also, the highest proportion, 672(52.5%) of women had deliveries at home and almost a similar proportion of 719 (56.1%) women did not have any health check within the six weeks of postpartum (Table 1).

### Results from survival analysis

Among all the respondents, 59.5% (95% CI: 56.8%–62.2%) had started using any modern contraceptive method after the last birth while the remaining were right- censored as of the time of the survey. Contraceptive users have contributed 11,737 months (978 women-years) of follow up, with an average follow-up time of 9.16 (95% CI = 8.96–9.37) months (**Fig 1**).

Our findings revealed that only 11.1 percent of the postpartum women surveyed started to use a contraceptive method by the second month after childbirth. The proportion of users then increased steadily over the months reaching 25.9%, 37.7%, and 59.5% at 6, 9, and 12 months, respectively.

The illustrations in Fig 2 also provide insights into the features of the differences in the Kaplan-Meier survival curves by selected maternal characteristics. Clearly, the overall estimated survivor function showed that mothers started taking modern contraception after the

**Table 1. Background characteristics of postpartum women in West Gojjam Zone, Northwest Ethiopia, 2018(n = 1281).**

| Variables | Survival Status | | |
| --- | --- | --- | --- |
| | Failures (contraceptive users) n = 762 | Censored (Nonusers) n = 519 | Total |
| **Residence** | | | |
| Rural | 551(72.3) | 427(82.3) | 978(76.3) |
| Urban | 211(27.7) | 92(17.7) | 303(23.7) |
| **Age of the women** | | | |
| 15–24 years | 165(21.7) | 61(11.8) | 226(17.6) |
| 25–34 years | 408(53.5) | 266(51.2) | 674(52.7) |
| > = 35 years | 189(24.8) | 192(37.0) | 381(29.7) |
| **Education status of women** | | | |
| No education | 409(53.7) | 415(79.9) | 824(64.3) |
| Primary education | 221(29.0) | 85(16.4) | 306(23.9) |
| Secondary and above | 132(17.3) | 19(3.7) | 151(11.8) |
| **Education status of husbands(n = 1213)** | | | |
| No education | 354(47.2) | 302(65.2) | 656(54.1) |
| Primary education | 246(32.8) | 133(28.7) | 379(31.2) |
| Secondary and above | 150(20.0) | 28(6.1) | 178(14.7) |
| **Wealth status** | | | |
| Poor | 349(45.8) | 243(46.8) | 592(46.2) |
| Middle | 157(20.6) | 113(21.8) | 270(21.1) |
| Rich | 256(33.6) | 163(31.4) | 419(32.7) |
| **Birth order of the last child** | | | |
| 1 | 212(27.8) | 76(14.7) | 288(22.5) |
| 2–4 | 392(51.5) | 269(51.8) | 661(51.6) |
| 5+ | 158(20.7) | 174(33.5) | 332(25.9) |
| **Family size** | | | |
| 1–3 | 212(27.8) | 98(18.9) | 310(24.2) |
| 4–6 | 406(53.3) | 263(50.7) | 669(52.2) |
| > = 7 | 144(18.9) | 158(30.4) | 302(23.6) |
| **Interval between the preceding non-first births(n = 993)** | | | |
| < 24 months | 23(4.2) | 23(5.2) | 46(4.6) |
| 24–33 months | 123(22.4) | 122(27.5) | 245(24.7) |
| 34–59 months | 358(65.2) | 272(61.2) | 630(63.4) |
| > = 60 months | 45(8.2) | 27(6.1) | 72(7.3) |
| **Intendedness of the last pregnancy** | | | |
| Intended | 667(87.5) | 395(76.1) | 1062(82.9) |
| Unintended | 95(12.5) | 124(23.9) | 219(17.1) |
| **Menses resumed after last birth** | | | |
| Yes | 605(79.4) | 165(31.8) | 770(60.1) |
| No | 157(20.6) | 354(68.2) | 511(39.9) |
| **Attended ANC (4+)** | | | |
| Yes | 390(51.2) | 121(23.3) | 511(39.9) |
| No | 372(48.8) | 398(76.7) | 770(60.1) |
| **Initiation time of first ANC after conception (n = 898)** | | | |
| First trimester | 121(20.0) | 13(4.4) | 134(14.9) |
| Second trimester | 409(67.7) | 206(70.1) | 615(68.5) |
| Third trimester | 74(12.3) | 75(25.5) | 149(16.6) |

*(Continued)*

**Table 1.** (Continued)

| Variables | Survival Status | | |
| --- | --- | --- | --- |
| | Failures (contraceptive users) n = 762 | Censored (Nonusers) n = 519 | Total |
| **Place of delivery** | | | |
| Healthcare facility | 482(63.3) | 127(24.5) | 609(47.5) |
| Home | 280(36.7) | 392(75.5) | 672(52.5) |
| **Mode of delivery(n = 615)** | | | |
| SVD | 392(80.7) | 129(86.6) | 501 (81.5) |
| Others(C/S, Assisted) | 94(19.3) | 20(13.4) | 114(18.5) |
| **At least one PNC within 6weeks after birth** | | | |
| Yes | 457(60.0) | 105(20.2) | 562(43.9) |
| No | 305(40.0) | 414(79.8) | 719(56.1) |
| **Early PNC (within 48–72 hours after birth)** | | | |
| Yes | 172(22.6) | 20(3.9) | 192(15.0) |
| No | 590(77.4) | 499(96.1) | 1089(85.0) |

SVD = Spontaneous Vaginal Delivery; C/S = Caesarean Section; Assisted delivery includes vacuum and forceps deliveries

2nd month of the last birth. It is also evident that the survival curves are substantially different among women whose first antenatal visit was during the first trimester, during the second trimester, or started in the third trimester. Similarly, the curves differ for various educational categories of women, the number of antenatal follow-ups, place of residence, and history of menstrual resumption after birth during the first12 months following the last childbirth (Fig 2).

Before we fit the final model, the observed difference in survival experiences in different groups was also assessed using the log-rank test. The variables considered include maternal age, maternal and paternal educational attainment, frequency and timing of first ANC follow up, type of delivery, postnatal care within the first two days after birth, PHC to population ratio, family size, and household wealth index.

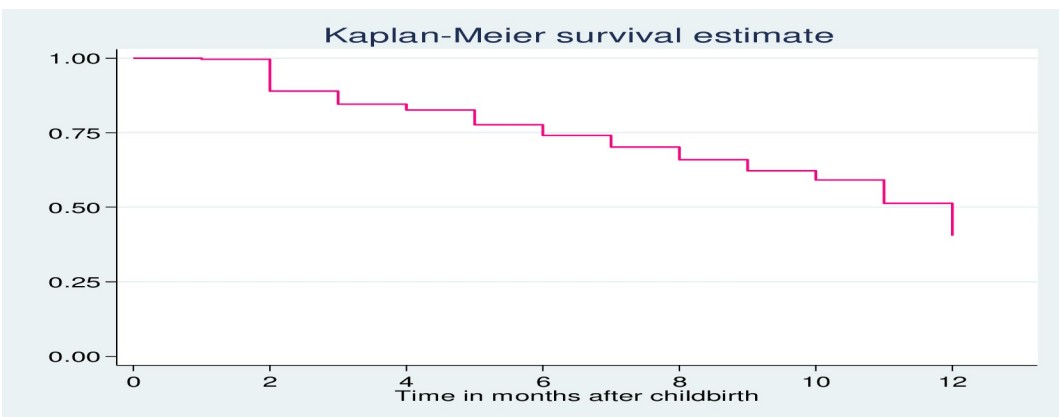

**Fig 1. Kaplan Meier survival function curve showing time to modern contraceptive use after birth among reproductive-age women in West Gojjam Zone, Northwest Ethiopia, and June 2018.**

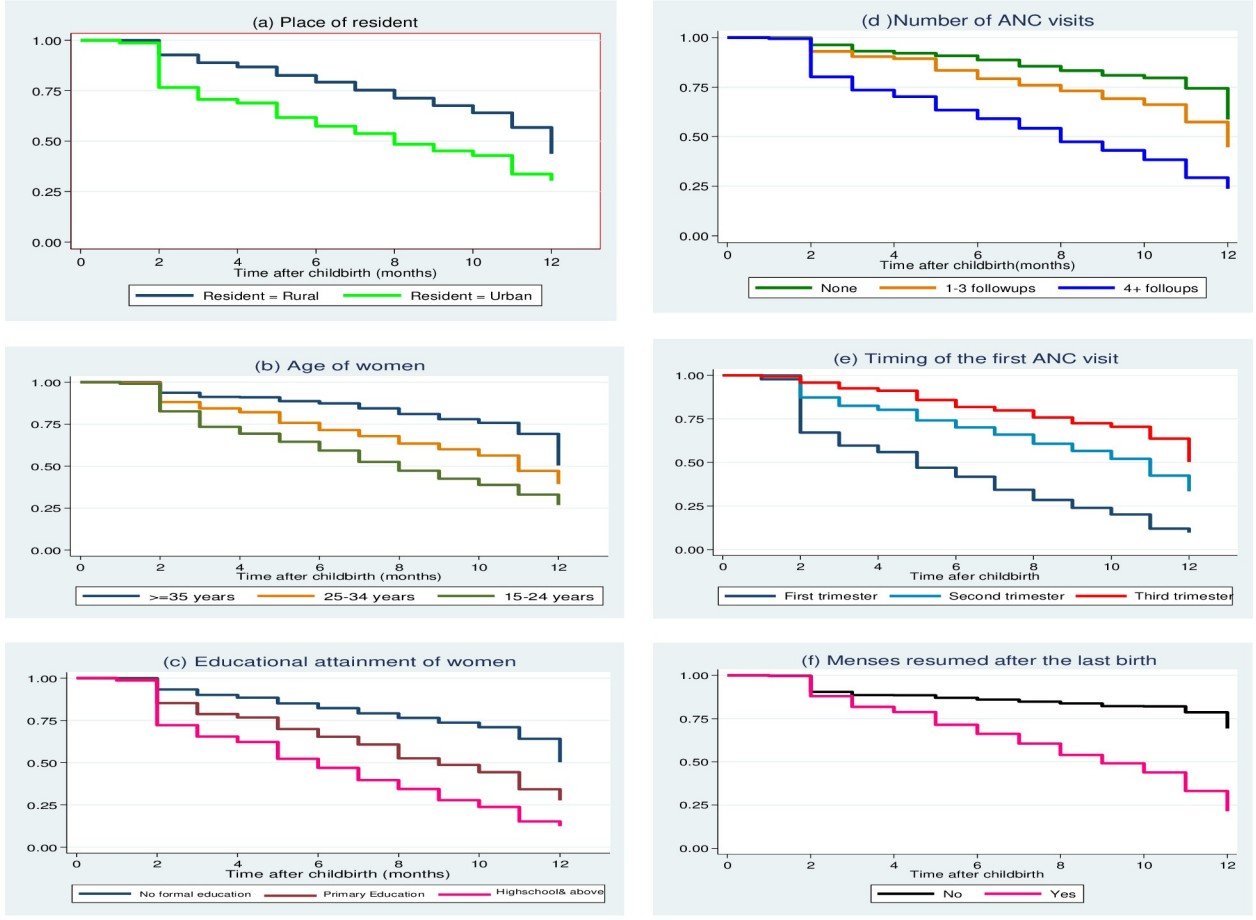

**Fig 2. Kaplan Meier estimate curve of postpartum family planning use within 12 months from childbirth by selected characteristics of women, Northwest Ethiopia, June 2018.**

The log-rank test result revealed that each of the covariates, except for facility to population ratio, has a significant Wald test when using α = 20%. However, we kept facility to population ratio in the final model as access to the healthcare facility was considered an important variable in different literature. Moreover, the sample size is sufficient to accommodate more predictors.

Then, all the covariates that were selected in the log-rank test at a 20% level of significance were fitted in the parametric shared frailty models of exponential, Weibull, log-logistic, and log-normal distributions by using cluster (kebele) as frailty term. The shared frailty model

**Table 2. Comparison of fitness of different parametric frailty models based on the Akaike information criteria, June 2018.**

| Baseline distribution | Frailty distribution | Log-likelihood | K | c | AIC Value |
|---|---|---|---|---|---|
| Exponential | Gamma | -1404 | 10 | 1 | 2829 |
| Weibull | Gamma | -603 | 10 | 2 | 1229 |
| Log-logistic | Gamma | -1343 | 10 | 2 | 2711 |
| Log-normal | Gamma | -1334 | 10 | 2 | 2691 |

AIC = -2lnL + 2(k +c), where c is the number of model-specific distributional parameters & k is the number of model covariates.

with the Weibull baseline hazard function had the smallest AIC value (Table 2) than the other frailty models, and hence was selected to describe time-to-postpartum contraception data. The AIC values for all the parametric frailty models are summarized in Table 2 below.

**Tests of Unobserved Heterogeneity.** The effect of clustering (unobserved heterogeneity) between the clusters (Kebeles) was tested using the likelihood ratio test (LRT). The results of this test revealed that the variance of the random effect was significantly greater than zero (θ = 0.06; p value< 0.05) in the weibull baseline frailty model, suggesting the unescapable role of the unmeasured cluster effects in the model. In addition, Kendall's tau (τ) value of 0.03, suggests a positive correlation between times to contraception within the clusters (Table 3).

**Goodness of fit of the final model.** We used the Cox-Snell residuals plot to check the overall goodness of fit for the final model. As depicted in Fig 3 below, the plot of the residuals of the fitted model is fairly closer to the $45^0$ straight line of the origin with slight variability in the right-hand tail, indicating that this model had a better fit to the data. Note that some variability about the 45˚ line is expected even with well-fitted survival models, particularly in the right-hand tail, because of the reduced effective sample caused by prior failures and censoring [25] (**Fig 3**).

## Multivariable survival analysis result

After controlling for other factors, the use of maternal health care services, and educational status of the women were found to be the sole predictors of postpartum modern contraceptive methods use (Table 3).

Our findings show that the hazard of postpartum modern contraceptive use was about 59% (aHR = 1.59; 95% CI: 1.22–2.06) higher for women who had a history of at least four antenatal visits compared with those who had less visits. Similarly, women who made their first antenatal visit within the first trimester had a two folds (aHR = 2.03; 95% CI: 1.28–3.21) risk of postpartum modern contraceptive use compared with those who first appeared in the last trimester.

The risk of starting modern contraceptive in the extended postpartum period was significantly higher (aHR = 1.39; 95% CI: 1.12–1.73) for women who received postnatal care within the first three days after birth compared with women who initiated postpartum care after three days.

Furthermore, in this study maternal education was significantly associated with the risk of using modern contraception in the postpartum period. When women completed primary education, their risk of using a modern method of contraceptive significantly increased by 1.30 times (aHR = 1.30; 95% CI: 1.02 1.66) compared with women who did not attend any formal education.

However, in this multivariable analysis, no statistical difference was observed in contraceptive uptake between women of different wealth status, area of residence, mode of delivery, and the number of health facilities per total population (facility density) in the district (Table 3).

## Discussion

This study has investigated the association of socioeconomic, demographic, and environmental factors with the likelihood of postpartum contraceptive use by accounting for gamma- distributed shared frailties at cluster-level.

In this study, we used kebele as a clustering (frailty) effect on modeling the determinants of time-to-contraception after birth. The statistically significant effect of the frailty terms between different clusters revealed that the observed covariates included in the analysis were not able to account for all the variability in women's survivorship. We postulate this variability to be the combined effect of various factors that cannot be easily measured or observed at community

**Table 3. Results of Multivariable analysis of time-to-contraceptive after childbirth by women aged 15–49 who had their most recent birth within 12 months preceding the survey, West Gojjam Zone, Ethiopia, June 2018.**

| | Hazard Ratio (95% CI) | | | |
|---|---|---|---|---|
| Variables | Unadjusted HR | 95% CI | aHR | 95% CI |
| **Residence** | | | | |
| Rural | 1.00 | - - - - - - - - - | 1.00 | . . . . . . |
| Urban | 1.77 | (1.46, 2.15) | 0.98 | (0.74, 1.32) |
| **Age of women** | 0.95 | (0.94, 0.97) | 0.99 | (0.97, 1.02) |
| **Educational status of women** | | | | |
| No education | 1.00 | - - - - - - - - | 1.00 | - - - - - |
| Primary education | 1.95 | (1.66, 2.30) | 1.298 | (1.02, 1.66)** |
| At least secondary | 3.37 | (2.75, 4.12) | 1.344 | (0.98, 1.84) |
| **Family size** | 0.88 | (0.84, 0.91) | 0.99 | (0.91, 1.09) |
| **Wealth status** | | | | |
| Poor | 1.00 | - - - | 1.00 | - - - |
| Medium | 1.03 | (0.85, 1.25) | 0.96 | (0.73, 1.25) |
| Rich | 0.99 | (0.84, 1.18) | 1.06 | (0.85, 1.32) |
| **Population per PHC** | | | | |
| >25,000population | 1.00 | - - - - - - - - | 1.00 | - - - - |
| < = 25,000 population | 1.01 | (0.71, 1.44) | 1.42 | (0.94, 2.12) |
| **Attended 4+ANC visits** | | | | |
| No | 1.00 | _____ | 1.00 | _____ |
| Yes | 2.48 | (2.13, 2.88) | 1.59 | (1.22, 2.06)** |
| **Timing of the first ANC** | | | | |
| Third trimester | 1.00 | - - - - - - - | 1.00 | - - - - - - |
| Second trimester | 1.67 | (1.30, 2.15) | 1.43 | (0.97, 2.12) |
| First trimester | 4.49 | (3.32, 6.08) | 2.03 | (1.28, 3.21)** |
| **Mode of delivery** | | | | |
| Non SVD (c/s, instrumental) | 1.00 | - - - - - | 1.00 | - - - |
| SVD | 0.79 | (0.63, 0.99) | 0.81 | (0.62, 1.04) |
| **Early PNC care (within 2–3 days after birth)** | | | | |
| No | 1.00 | - - - - | 1.00 | - - - |
| Yes | 3.31 | (2.78, 3.96) | 1.39 | (1.12, 1.73)** |
| **θ value (theta)** | | | 0.06 | |
| **τ (tau Kendall)** | | | 0.03 | |
| **Log-likelihood(LL)** | | | -603 | |

** Significant at 0.05level of significance, HR = Hazard Ratio, aHR = Adjusted Hazard Ratio; CI = Confidence Interval; C/S = Caesarian Section; SVD = Spontaneous Vaginal Delivery

LRT = Likelihood ratio test of theta ($\theta$) = 0 at chi-squares with 0 and 1 degrees of freedom; Kendall's tau ($\tau$) = $\theta/\theta+2$, where $\tau = \epsilon$ (0, 1). 95% Confidence interval

or health facility levels including culture and social norms[15]. The significant level of frailty terms in our study might have biased our results if we had not taken them into account [21]. Our interpretation, therefore, was based on the shared survival frailty model that accounts for the heterogeneity.

In this study, modern contraceptive practice during the extended postpartum period was found to be 59.5% (95% CI: 56.8–62.2%). The finding is comparable with the finding of the study done in Debre Tabor town, Northwest Ethiopia 63.0% (95% CI: 59%; 67.4%) [26]. This result was, however, higher than the 31.7% prevalence in Southern Ethiopia[20], 45.4% in

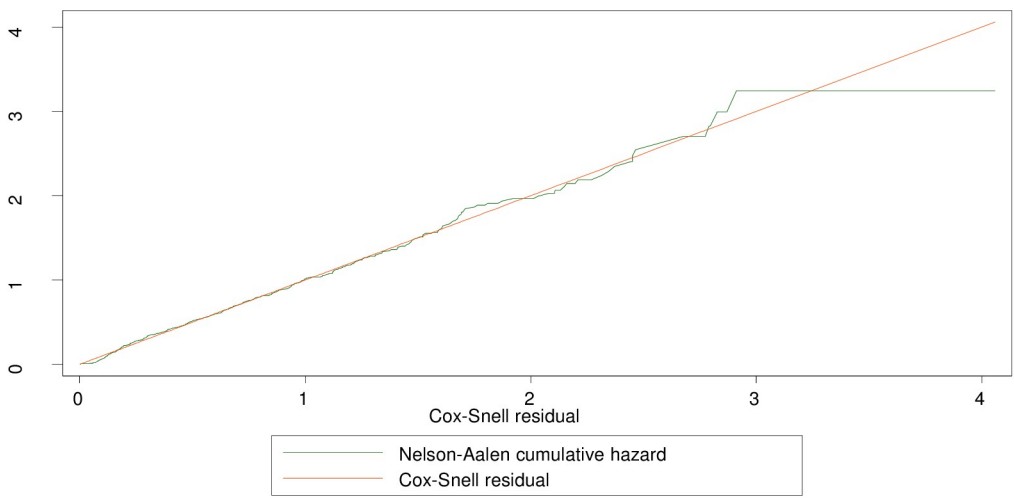

**Fig 3. Cumulative hazard of Cox-Snell residuals.**

western Ethiopia[27], and 29.3% in Northern Ethiopia[19], and lower when compared to the studies done in Hosana town (72.9%), Addis Ababa, Ethiopia (80.3%), and 86.3% of Kenya [28–30]. The low level of contraceptive use found in this study might reflect the over- reliance of lactating women on breastfeeding and menstruation status. It is evidenced that these group of women oftentimes do not realize that they are at risk of pregnancy when they are amenorrheic or breastfeeding [15, 20, 26]. Though breastfeeding is universal in Ethiopia, and exclusive breastfeeding up to 6 months after birth is an important contraceptive method which is highly recommended by the Ministry of Health of Ethiopia, the status of exclusive breastfeeding in the country is less than the global recommendations [31].

The main finding of this analysis is that women who started using modern contraceptive methods during the extended postpartum period were characterized by high coverage (four or more visits) and proper timing (first visit in first trimester) of antenatal care. Yet, the results of studies done in Ethiopia and elsewhere [10, 18, 28]showed that postpartum use of modern contraception was not affected by antenatal care. The variation could be attributed to the difference in the study design; whereas our study accounted for the hierarchical structure and tried to adjust for individual and community characteristics, the other studies were done using flat models that inherently assume the population to be homogeneous. It is evidenced that frailty models offer unobserved heterogeneity into models for survival data as random effects [21].

Nonetheless, the observed association between prenatal care and contraceptive use is not unique to this study and has been reported in earlier studies from Ethiopia [20, 27, 32], and other countries [8, 16]. These studies showed a dose-response type of relationship between antenatal care and postpartum contraception adoption; that is the likelihood of using postpartum contraception increased when women had frequent antenatal contacts. Our result also demonstrated a significant association between postpartum contraceptive use and early postnatal care, which is in agreement with other studies [20, 27].

The positive effects of maternity services on contraceptive uptake might be explained due to the effect of the counseling sessions and promotional efforts made during each visit. It has been indicated that each maternity services improve clients' relationships with health workers and their familiarity with the health care systems [33]. Besides, counseling and information

can help women avoid social barriers and, in turn, encourage them to use health services in the future [34]. Therefore, cognizant of the fact that only a few Ethiopian women have gotten antenatal and early postnatal services [3], there is a strong need to promote programs that target women who do not get these services as a strategy to promote postpartum modern contraceptive use.

There are inconsistent pieces of evidence in the correlation between women's education and postpartum contraception adoption. A study done in Northwest Ethiopia, for example, did not show any association between female education and postpartum contraceptive uptake [18]. On the other hand, in line with the previous researches[10, 15], the result of our study also confirms that educated women have a higher hazard of contraceptive use when compared to mothers with no formal education. Women's education could impact modern contraceptive uptake in different mechanisms: improving access to contraceptive alternatives, and helping them in understanding the health benefits of the available contraceptive commodities [35] might be among the possible reasons. Education might also improve the bargaining power of women to negotiate sex, and their ability to make their own decisions, including fertilities desires [36].

Results from various studies have found conflict of evidence on the link between household wealth status and the use of postpartum contraception; in some settings, it appears to be associated with contraceptive use; Hounton and colleagues, for example, reported financial constraint as a barrier to adopt postpartum contraception[10]. However, no statistically significant difference was observed in contraceptive uptake between women of different wealth status in our study, which is in line with the results of some other studies[15, 28].

This lack of variation in contraceptive use by wealth status in our study might be attributed to the introduction of healthcare financing reforms by the government of Ethiopia, which includes social and community based health insurance schemes, and charge free maternity services in public health facilities, among others[37]. Concerning this, Dzakpasu et al reported that poor women were unwilling to use the formal health sector if they must pay for maternal health services[38]. The Health Extension Program in Ethiopia, which brought family planning services to the community where they live, might be another reason for the lack of variation between rich and poor women. Health extension workers are deployed in pairs, two for every kebele, and affiliated with each kebele's health post to provide key health services at a community level, including family planning services since launched in 2003 [39].

Despite we tried to estimate unbiased parameter estimates after accounting for the frailty effect, the study results should also be interpreted in light of certain limitations. The reliability of this study depends on the mother's recall of past events regarding the processes of maternal health care and therefore may be subject to recall bias. In addition, the study focused merely on the health coverage of maternal services as main predictors, yet coverage alone might not be a warranty for postpartum contraceptive use if quality was insufficient[40].

## Conclusions

In conclusion, the use of postpartum modern contraception was low despite the provision of charge-free services in all public health facilities. Postpartum modern contraception use was associated with increased coverage of the key maternal services, particularly the antenatal and postnatal cares. The observed strong effect of antenatal and early postnatal services strengthens the argument that integrating the key maternity could enhance the use of postpartum modern contraception. Moreover, the significant level of variance of unobserved community effect also underscores the importance of disaggregated data for evidence-based policymaking and program designing in the study area in particular and the country in general.

## Supporting information

**S1 Table. Data collection tool (English).**
(PDF)

**S2 Table. Data collection tool (Amharic).**
(PDF)

## Acknowledgments

We would like to express our deepest gratitude to the Amhara regional health bureau, West Gojjam Zone Health Department, and district health offices. We also extend our acknowledgment to the data collectors, supervisors and study participants.

## Author Contributions

**Conceptualization:** Amanu Aragaw Emiru.

**Data curation:** Amanu Aragaw Emiru, Getu Degu Alene, Gurmesa Tura Debelew.

**Formal analysis:** Amanu Aragaw Emiru, Getu Degu Alene, Gurmesa Tura Debelew.

**Funding acquisition:** Amanu Aragaw Emiru, Getu Degu Alene, Gurmesa Tura Debelew.

**Investigation:** Amanu Aragaw Emiru, Getu Degu Alene, Gurmesa Tura Debelew.

**Methodology:** Amanu Aragaw Emiru, Getu Degu Alene, Gurmesa Tura Debelew.

**Project administration:** Amanu Aragaw Emiru, Getu Degu Alene, Gurmesa Tura Debelew.

**Resources:** Amanu Aragaw Emiru, Getu Degu Alene, Gurmesa Tura Debelew.

**Software:** Amanu Aragaw Emiru, Getu Degu Alene, Gurmesa Tura Debelew.

**Supervision:** Getu Degu Alene, Gurmesa Tura Debelew.

**Validation:** Amanu Aragaw Emiru, Getu Degu Alene, Gurmesa Tura Debelew.

**Visualization:** Amanu Aragaw Emiru, Getu Degu Alene, Gurmesa Tura Debelew.

**Writing – original draft:** Amanu Aragaw Emiru.

**Writing – review & editing:** Amanu Aragaw Emiru, Getu Degu Alene, Gurmesa Tura Debelew.

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
