## [Decision Letter · Decision Letter 0]

2 Jan 2020

PONE-D-19-27516

The role of maternal health care services as predictors of time to modern contraceptive use after childbirth in North West Ethiopia: Application of the shared frailty survival analysis

PLOS ONE

Dear Mr Emiru,

Thank you for submitting your manuscript to PLOS ONE. After careful consideration, we feel that it has merit but does not fully meet PLOS ONE’s publication criteria as it currently stands. Therefore, we invite you to submit a revised version of the manuscript that addresses the points raised during the review process.

We would appreciate receiving your revised manuscript by Feb 16 2020 11:59PM. To enhance the reproducibility of your results, we recommend that if applicable you deposit your laboratory protocols in protocols.io, where a protocol can be assigned its own identifier (DOI) such that it can be cited independently in the future. For instructions see: http://journals.plos.org/plosone/s/submission-guidelines#loc-laboratory-protocols

We look forward to receiving your revised manuscript.

Kind regards,

Kannan Navaneetham

Academic Editor

PLOS ONE

Journal Requirements:

2. Please include additional information regarding the survey or questionnaire used in the study and ensure that you have provided sufficient details that others could replicate the analyses. For instance, if you developed a questionnaire as part of this study and it is not under a copyright more restrictive than CC-BY, please include a copy, in both the original language and English, as Supporting Information. Also, you referred to pre-testing of this questionnaire but did not provide details of the nature or number of participants.

3. Please provide additional details regarding participant consent. In the ethics statement in the Methods and online submission information, please ensure that you have specified how oral consent was documented and witnessed. If your study included minors, state whether you obtained consent from parents or guardians. If the need for consent was waived by the ethics committee, please include this information.

Reviewers' comments:

Reviewer's Responses to Questions

**Comments to the Author**

1. Is the manuscript technically sound, and do the data support the conclusions?

Reviewer #1: Yes

Reviewer #2: Yes

2. Has the statistical analysis been performed appropriately and rigorously? 

Reviewer #1: Yes

Reviewer #2: Yes

3. Have the authors made all data underlying the findings in their manuscript fully available?

Reviewer #1: Yes

Reviewer #2: Yes

4. Is the manuscript presented in an intelligible fashion and written in standard English?

Reviewer #1: Yes

Reviewer #2: Yes

5. Review Comments to the Author

Reviewer #1: on line 193, the index for frailty term is not correct, basing on equation specification j index represent for individual and i for cluster. For this case frailty is taking into consideration clusters heterogeneity not individuals.

For tables, they should add percentage to other column as they did for total column.

Reviewer #2: The manuscript is related to the role of maternal health care services as predictors of time to modern contraceptive use after childbirth in North West Ethiopia using Gamma distribution for shared frailty survival analysis with different distributions. The manuscript is generally well written and clearly presented. However, some minor corrections should be done.

1- In Table 1 for each variable percentages of contraceptive users and nonusers are needed.

2- For table 3, for Log-likelihood and AIC values, decimals are not needed.

3- In Table 4, the number of decimal places must be the same.

4- Information of Fig1 with Table2 are the same. One of them should be deleted.

5- Please check baseline hazard function of Weibull distribution in Line192. What is the scale parameter of the Weibull distribution?

6- The interpretations of Multivariable Survival Analysis results should be modified. For example, what do you mean “The likelihood of initiating contraceptive use for women who had four or more antenatal visits was 59 percent (aHR =1.59; 95% CI: 1.22–2.06) higher than for women who had less number of visits.”

7- There is no for some sentences reference. For e.g. “Consequently, modeling time to contraceptive use ignoring the frailty terms may lead to biased estimates of parameters and their respective standard errors.”. Please check the whole manuscript again.

6. PLOS authors have the option to publish the peer review history of their article (what does this mean?). If published, this will include your full peer review and any attached files.

Reviewer #1: Yes: Semakula Muhammed

Reviewer #2: Yes: Prof. Hossein Mahjub

---

## [Author Response · Author response to Decision Letter 0]

13 Jan 2020

Response to Reviewers

Dear Editor,

We are very grateful for your letter and the opportunity to improve and resubmit our manuscript. We appreciate the time and effort that you and the reviewers have dedicated to providing your valuable comments and suggestions on our manuscript. The comments and suggestions provided have been immensely helpful. We have revised the manuscript based on the comments and suggestions.

Kindly find below our point-by-point responses to the reviewers’ comments. Unless otherwise stated, all line numbers and pages are in reference to the revised manuscript. 

Sincerely,

Amanu Aragaw Emiru

(On the behalf of Co-Authors)

Comments from Reviewer 1

Comment #1: on line 193, the index for frailty term is not correct, basing on equation specification j index represent for individual and i for cluster. For this case, frailty is taking into consideration clusters heterogeneity not individuals.

Response: Thank you for pointing this out; please consider this as a typo. This has now been corrected (page 9, line 193) 

Comment #2: For tables, they should add percentage to other column as they did for total column.

Response: Agree. We have incorporated percentages both for contraceptive user and for nonuser columns (page 11-12, line 223). 

Comments from Reviewer 2

Comment #1: In Table 1, for each variable percentages of contraceptive users and nonusers are needed. 

Response: Thank you for your suggestion. We have added percentages for both contraceptive user and nonuser columns

Comment #2: For table 3, for Log-likelihood and AIC values, decimals are not needed.

 Response: Decimals are removed as per your suggestion, thank you.

Comment #3: In Table 4, the number of decimal places must be the same.

Response: Done. 

Comment #4: Information of Fig1 with Table2 are the same. One of them should be deleted.

Response: We agree, we have removed table 2 from the manuscript 

Comment #5: Please check baseline hazard function of Weibull distribution in Line192. What is the scale parameter of the Weibull distribution?

Response: We agree that the symbols was not clear as it stands. We have slightly modified the hazard function to ease readers understand the model. In the revised version, Zi exp (β’xij) denotes the scale parameter and ρ the shape parameter. 

Comment #6: The interpretations of Multivariable Survival Analysis results should be modified. For example, what do you mean “The likelihood of initiating contraceptive use for women who had four or more antenatal visits was 59 percent (aHR =1.59; 95% CI: 1.22–2.06) higher than for women who had less number of visits.”

Response: We are grateful for this comment. We tried to revise this section as per your comment. The revised section is highlighted on line 288-299 (page 15)

Comment #7: There is no for some sentences reference. For e.g. “Consequently, modeling time to contraceptive use ignoring the frailty terms may lead to biased estimates of parameters and their respective standard errors.” Please check the whole manuscript again.

Response: Thanks for this comment and suggestion. We have carefully read the manuscript and have added references as advised. The references considered are highlighted in the revised version.

---

## [Editor Report · Decision Letter 1]

22 Jan 2020

The role of maternal health care services as predictors of time to modern contraceptive use after childbirth in North West Ethiopia: Application of the shared frailty survival analysis

PONE-D-19-27516R1

Dear Dr. Emiru,

We are pleased to inform you that your manuscript has been judged scientifically suitable for publication and will be formally accepted for publication once it complies with all outstanding technical requirements.

With kind regards,

Kannan Navaneetham

Academic Editor

PLOS ONE
---

## [Editor Report · Acceptance letter]

28 Jan 2020

PONE-D-19-27516R1 

The role of maternal health care services as predictors of time to modern contraceptive use after childbirth in Northwest Ethiopia: Application of the shared frailty survival analysis 

Dear Dr. Emiru:

I am pleased to inform you that your manuscript has been deemed suitable for publication in PLOS ONE. Congratulations! Your manuscript is now with our production department. 

With kind regards,

on behalf of

Professor Kannan Navaneetham 

Academic Editor

PLOS ONE